# What the Gut Tells the Brain—Is There a Link between Microbiota and Huntington’s Disease?

**DOI:** 10.3390/ijms24054477

**Published:** 2023-02-24

**Authors:** Dorota Wronka, Anna Karlik, Julia O. Misiorek, Lukasz Przybyl

**Affiliations:** 1Laboratory of Mammalian Model Organisms, Institute of Bioorganic Chemistry Polish Academy of Sciences, Noskowskiego 12/14, 61-704 Poznan, Poland; 2Department of Molecular Neurooncology, Institute of Bioorganic Chemistry Polish Academy of Sciences, Noskowskiego 12/14, 61-704 Poznan, Poland

**Keywords:** Huntington’s disease, neurodegeneration, gastrointestinal microbiome, gut-brain axis, dysbiosis, immune

## Abstract

The human intestinal microbiota is a diverse and dynamic microenvironment that forms a complex, bi-directional relationship with the host. The microbiome takes part in the digestion of food and the generation of crucial nutrients such as short chain fatty acids (SCFA), but is also impacts the host’s metabolism, immune system, and even brain functions. Due to its indispensable role, microbiota has been implicated in both the maintenance of health and the pathogenesis of many diseases. Dysbiosis in the gut microbiota has already been implicated in many neurodegenerative diseases such as Parkinson’s disease (PD) and Alzheimer’s disease (AD). However, not much is known about the microbiome composition and its interactions in Huntington’s disease (HD). This dominantly heritable, incurable neurodegenerative disease is caused by the expansion of CAG trinucleotide repeats in the huntingtin gene (*HTT*). As a result, toxic RNA and mutant protein (mHTT), rich in polyglutamine (polyQ), accumulate particularly in the brain, leading to its impaired functions. Interestingly, recent studies indicated that mHTT is also widely expressed in the intestines and could possibly interact with the microbiota, affecting the progression of HD. Several studies have aimed so far to screen the microbiota composition in mouse models of HD and find out whether observed microbiome dysbiosis could affect the functions of the HD brain. This review summarizes ongoing research in the HD field and highlights the essential role of the intestine-brain axis in HD pathogenesis and progression. The review also puts a strong emphasis on indicating microbiome composition as a future target in the urgently needed therapy for this still incurable disease.

## 1. Introduction

### 1.1. Intestinal Microbiome

The intestinal microbiome is the largest and most active group of microorganisms in the human body. It plays an essential role in health and disease, but due to its complexity, it is challenging to elucidate the specific interactions between the bacterial species and the impact on host metabolism. The large intestine (colon) is the main place inhabited by microbiota. It is built up by several tissue types, including lumen-facing colonocytes that form the inner epithelial layer. A healthy microbiome is advantageous to the host due to its ability to digest various large molecules, like long plant-derived polysaccharides, into smaller nutrients, like short chain fatty acids (SCFA), that can be absorbed and utilized by the host. It also produces various other molecules, such as amino acids, vitamins, and neurotransmitters, that contribute to the host’s health [1,2]. Over 1000 different bacterial species colonize the human gut, the vast majority of which have yet to be functionally characterized. The microbiota composition is dynamic and influenced by a variety of environmental factors such as diet, physical activity, host genetics, age, and antibiotic treatment, all of which contribute to the great diversity observed in healthy individuals. It is thus a challenge to accurately characterize a healthy microbiome [3]. We took a closer look at several large-scale studies that point to the genera *Bacteroides* and *Clostridium* as being the most prevalent, with *Clostridium* being less abundant than *Bacteroides* in the human intestine. Several genera, including *Bifidobacterium*, *Eubacterium*, *Lactobacillus*, *Streptococcus,* and *Escherichia,* were also present but in much lower abundance [3]. Determining a clear definition of a “healthy” microbiome is challenging, and many various factors need to be considered. The microbiome composition is dependent on a multitude of factors that may seem insignificant at first glance. In 2010, studies conducted by the MetaHIT consortium made an attempt to quantify microbiome diversity. According to the obtained results, there are 3.3 million non-redundant genes in the human gut microbiome [4], however, it had been known until early 2000s that the human genome consists of about 22,000 genes [5]. Further research confirms that the diversity of the microbiome is enormous between individuals and can differ by up to 90% in terms of microbiome localization (e.g., those found on the hands vs. those present in the gut) [6,7]. These findings drive scientists and physicians towards developing a highly personalized treatment plan. The profile and microbiota composition changes with the host’s lifespan, starting from embryos which were thought to be sterile till now. The microbiota colonizes newborns’ intestines, but studies have also revealed the microbiome’s presence in semen, placenta, amniotic fluid, umbilical cord blood, and meconium [8]. Moreover, factors such as delivery and feeding methods are essential for microbiota composition in infants and adults. Further, when children start to ingest solid food, their intestinal microbiome becomes more diversified, and during puberty, the release of sex hormones also contributes to microbiome maturation [9]. Next, diversification of the microbiome occurs naturally with the physiological development of the organism, i.e., the increase in length and volume of the intestines provides the microbiome with appropriate niches. Numerous studies indicate that there is a correlation between aging and microbiome composition. In 2011, a pioneering study was conducted to compare the composition of the microbiome in fecal samples from people aged 64 to 102 (study group) and young adults with an average age of 36 (control group). The results showed that the “core” microbiome—defined as the specific species found in the microbiome of at least 50% of study participants—was significantly different between the groups [10,11]. So far, the main function of the intestinal microbiome has been identified as maintaining body homeostasis. Researchers emphasize that despite the fact that technological progress is at a high level, the individual composition of the microbiome, functional characteristics, or interactions between the host and microbes have not yet been established [12]. Data collected by the Human Microbiome Project [13,14] and MetaHIT [4,15] report that 2776 species of prokaryotic microorganisms isolated from human feces have been identified (data for 2019) [16]. They have been classified into 11 different phyla, including *Proteobacteria, Firmicutes, Actinobacteria,* and *Bacteroidetes*, which make up over 90% of the microbiome, [15,17,18], while *Fusobacteria* and *Verrucomicrobia* are present in trace amounts [19]. As mentioned earlier, microbiota are essential for the proper function and homeostasis of the intestines. Interactions between gut colonocytes, immune cells, and microbiota are heavily involved in shaping the immune response throughout the body [20]. In support of this, gut microbiota transplants from healthy individuals have been found to alleviate symptoms and reduce inflammation in disorders like ulcerative colitis, irritable bowel syndrome (IBS), and hepatic encephalopathy [21,22]. 

### 1.2. Short Chain Fatty Acid Production and Their Importance

Key end products of microbial fermentation in the large intestine are short chain fatty acids. They are saturated carboxylic acids containing less than six carbons in their chain structure. The main sources of SCFAs are dietary macromolecules, especially fiber-rich plant-derived polysaccharides that are indigestible to humans due to the lack of enzymes required for breaking the glycosidic bonds. Thus, they are available to microbes in the intestinal lumen, which ferment them and make them available to the host. SCFAs are transported into the colonic epithelial cells by solute transporters or by simple diffusion across the membranes [23]. 95% of the total SCFAs in the human gut are acetate, propionate, and butyrate, and their levels are largely dependent on the diet and the amount of fiber, which affect the microbiota composition. The main species involved in the production of acetate are *Akkermansia muciniphila, Bacteroidetes* spp., and *Prevetolla* spp. Propionate is mostly produced by *Bacteroidetes* and *Firmicutes*, with the latter also producing butyrate. SCFAs are an important energy source for colonocytes and hepatocytes, but they also enter the systemic circulation and act as signaling molecules to exert a variety of regulatory functions. The presence of SCFAs is closely linked to gut integrity, not only through increased expression of tight junction (TJ) proteins but also through modulation of the host immune system. They act as ligands for G-protein-coupled receptors (GPR), their main targets being GPR43 and GPR41, also called free fatty acid receptor-2 (FFAR2) and free fatty acid receptor-3 (FFAR3), respectively. It has also been reported that butyrate can interact with GPR109/HCA2 (hydroxycarboxylic acid receptor 2). These receptors are involved in the glucose metabolism, lipid regulation, and gut homeostasis, as well as being expressed on immune cells, where they can influence the inflammation. Indeed, acetate has been implicated in resolving enteritis through GPR43 signaling [24]. Propionate, butyrate, and valerate can influence gene transcription by inhibiting histone deacetylase (HDAC) and thus making chromatin more accessible to transcription factors. Butyrate has been shown to be a potent suppressor of CD4^+^ T cell activation, acting through GPR43 and HDAC inhibition to decrease proliferation and production of proinflammatory cytokines (IFN-γ, IL-17) [25,26]. Studies show that butyrate-mediated inhibition of class II HDAC in the gut CD4^+^ T cells epigenetically induces the transcription of genes responsible for regulatory T cell (Treg) function [27]. There are many examples of the anti-inflammatory roles of SCFAs, but some studies report a dual effect, inducing both Treg and cytotoxic effector T cells, which points out the need for further studies [23].

Importantly, SCFAs can also cross the blood-brain barrier and affect the brain, which renders them as a potential target in neuroinflammatory diseases [20]. Supplementation of sodium butyrate has been tested on the R6/2 mouse model of HD, yielding positive results. When compared to untreated controls, the supplemented group showed improved motor performance, increased brain weight, and decreased striatal neuronal atrophy. However, sodium butyrate supplementation had no effect on the formation of mutant huntingtin (mHTT) aggregates or weight loss [28]. The study conducted on the YAC128 mouse HD model has also shown a beneficial effect of sodium butyrate supplementation, as the treated group displayed improved learning and motor skills, as well as improved cortical energy levels and increased histone 3 acetylation, suggesting that butyrate acting as an HDAC inhibitor can improve mitochondrial and transcriptional dysfunctions present in HD [29].

### 1.3. Tryptophan Metabolism

Tryptophan is an essential amino acid, since in mammals it is mainly derived from diet and used for protein synthesis or converted through two main pathways: serotonin or kynurenine. In the body, there are two pools of serotonin: the brain and the gut. In the brain, serotonin is synthesized in the midbrain by neurons of the raphe nucleus, although the vast majority of serotonin is produced in the gut and can impact the brain through the stimulation of the vagus nerve. Other microbial metabolites, such as butyrate, can also impact serotonin production by stimulating the activity of the tryptophan hydroxylase 1 (TPH1) enzyme. The serotonin pathway can also lead to the synthesis of melatonin, which regulates the biological rhythm and can have antioxidant and anti-inflammatory effects [30]. 

The kynurenine pathway utilizes the vast majority of available tryptophan and leads to the synthesis of NAD^+^, which is essential for the proper functioning of the cells. There are two enzymes responsible for the conversion of tryptophan into kynurenine: IDO1 and IDO2. The IDO1 enzyme has been implicated as a key molecule regulating the host-microbiome symbiotic relationship and immune responses. L-kynurenine acts as a ligand for the aryl hydrocarbon receptor (AhR), which is expressed in lymphoid tissues and has been linked to promoting Treg development in the periphery, thus stimulating homeostasis and immune tolerance. AhR signaling is also responsible for promoting IL-22 expression in gut-resident type 3 innate lymphoid cells (ILC3) [31]. There are two major metabolites synthesized along this pathway that have neuroactive properties: kynurenic acid (KYNA) and quinolinic acid (QUIN). KYNA has a neuroprotective function and is mainly produced by astrocytes, while QUIN has neurotoxic effects and is synthesized by microglia. The presence of IFN-γ and a proinflammatory environment has been found to promote QUIN production and skew the balance towards neurotoxicity. 

Additionally, the gut microbiome can metabolize tryptophan along the indole pathway. *Escherichia coli*, *Clostridium* spp., and *Bacteroides* spp. are known to utilize this pathway. About 5% of ingested tryptophan is used by microbes for a variety of physiological processes, like biofilm formation, drug resistance, virulence, and others, which are required for the maintenance of a variable microbial community, but indole and its derivatives also influence the host [30,32]. Similar to kynurenine, several indole derivatives can act as ligands for AhR and have been linked to promoting IL-22 expression. A study has shown that regulation of gut IL-22 expression by indole-3-aldehyde allows for the survival of a varied microbial community while providing resistance to opportunistic fungi (*C. albicans*) infection [31].

### 1.4. Gut-Brain Axis

The gut-brain axis is the main link between the digestive tract and the central nervous system (CNS). It is a specific two-way communication system consisting of neural pathways such as the enteric nervous system (ENS), the sympathetic and spinal vagus nerves, and the humoral pathways involving cytokines, hormones, and neuropeptides [33]. The factors regulating the work of the axis include cortisol, SCFAs, neurotransmitters, neuromodulators, and the intestinal microbiota, which has been recognized relatively recently and is still gaining popularity. For a long time, the gut-brain axis has been known to play a role in maintaining homeostasis in the body. Disturbances of the brain-gut axis are believed to lead to systemic disorders, such as dysregulation of the intestinal system and CNS disorders, e.g., depression [34,35]. The direct impact of the microbiome on the CNS is still poorly understood. The gut microbiome is known to produce neurotransmitters such as gamma-aminobutyric acid (GABA), histamine, dopamine, norepinephrine, and serotonin, as well as most likely other neuroactive molecules [16]. The ENS is the internal nervous system of the gastrointestinal tract, where neurons organized in microarrays enable modulation of gastrointestinal function independently from the CNS, although the systems are interconnected and interact with each other [36]. This combination is also believed to allow the neurodegenerative diseases to progress. In 80% of individuals affected by Parkinson’s disease, the symptoms of neurodegeneration were preceded by digestive system symptoms. It has been suggested that alpha-synucleopathy of the gastrointestinal nervous system is an early indicator of Parkinson’s disease. The regular expression of the *APP* gene in the ENS indicates that it is also involved in the pathogenesis of Alzheimer’s disease [37,38]. 

## 2. Neurodegenerative Disease Characterization and Link to Microbiome

### 2.1. Parkinson’s Disease and Alzheimer’s Disease

Two of the most prevalent neurodegenerative diseases are Parkinson’s disease (PD) and Alzheimer’s disease (AD), with the latter being more common. They are both progressive and associated with advanced age, but their exact causes are not fully understood, although it is believed that a combination of both genetic and environmental factors play a role in their development and progression. AD is mostly associated with memory loss, disorientation, and behavioral issues. In the brain, there is a progressive loss of neurons and the formation of amyloid plaques and neurofibrillary tangles originating from the amyloid-beta (Aβ) precursor protein (APP). PD is characterized by abnormal accumulation and aggregation of alpha-synuclein in the form of Lewy bodies and loss of dopaminergic neurons in the substantia nigra, which causes dopamine deficiency. The most common motor symptoms are tremors, stiffness, bradykinesia, and loss of coordination, with accompanying cognitive disorders such as depression, anxiety, and apathy [39,40]. 

The composition of the intestinal microbiota is not only important for maintaining the proper health of the body but can also affect the physiological, behavioral, and cognitive functions of the brain. There is ample evidence for differences in the microbiome between healthy individuals and PD patients. Patients suffering from PD were characterized by a reduced presence of *Prevotellaceae* bacteria and an increased number of *Enterobacteriaceae* bacteria. Currently, it is difficult to clearly define the role of SCFAs in the pathogenesis of neurodegenerative diseases. However, the vast majority of publications indicate pathological SCFA activity in PD patients. Studies in mice overexpressing alpha-synuclein demonstrate the effect of a microbial-free environment on the elimination of the PD phenotype, and oral feeding of SCFAs to the same mice restores the neuropathology associated with PD. Counterintuitively, SCFA administration to patients increases motor dysfunction and inflammation [41,42,43]. According to a study published in 2019, bacteria from the *Prevotellaceae* family have been found to provide high levels of health-promoting neuroactive SCFAs, which in turn contribute to a healthy environment in the gut [44]. Decreased *Prevotella* abundance has also been linked to multiple sclerosis (MS), type 1 diabetes, and autism spectrum disorders. Furthermore, the presence of *Prevotella* is significantly influenced by a plant-based diet. Increased abundance of *Lactobacillus* has been associated with type 2 diabetes and constipation, suggesting that the prognostic value of *Lactobacillus* is not specific to PD. Multiple bacterial taxa have been reported to be altered in individuals with PD. Potential interactions between them indicate that the effects of altered gut microbiota in PD may be the result of many complex cascades of events within the entire gut microbiota as well as relationships with the host [45].

Recent results suggest a strong link between the pathogenesis of AD and intestinal microbiota dysfunctions. Studies conducted on the ADLPAPT mouse model of AD show that changes in the composition of the intestinal microflora led to a loss of intestinal epithelial integrity, which in turn caused systemic inflammation. Intestinal abnormalities coincided with Aβ deposition, Tau protein pathology, progressive gliosis, and cognitive impairment in the animals. It was also noted that the transplantation of microbiota from healthy animals into animals suffering from AD significantly attenuated the progression of AD pathogenesis [46]. A number of studies indicate significant changes in the composition of the gut microbiota during the course of AD. There was an increase in *Firmicutes/Bacteroidetes* and a decrease in *Actinobacteria* and SCFA-producing bacteria in AD mice [47,48]. A large body of research supports the idea that the gut microbiome in mouse models of AD is less diverse than in wild type (WT) mice [48,49,50,51,52]. Some association has also been noted between the presence of butyrate- and lactate-producing bacteria. Furthermore, a decrease in the number of butyrate-producing *Faecalibacterium* and an increase in the number of lactate-producing bacteria of the *Bifidobacterium* family were found using the sequencing of 16S rRNA from stool samples [50]. Metagenomic studies have proven the relationship between *Lachnospiraceae* and type 2 diabetes. The aforementioned family of bacteria contributes to the development of diabetes, which, along with insulin resistance, is one of the risk factors for AD [53,54,55]. Functional studies show that *Pseudomonas aeruginosa* infection can increase endothelial Tau phosphorylation and permeability, a common pathophysiological mechanism in the genesis of Alzheimer’s disease [56,57]. To date, little has been established about the interactions between pathogenic and non-pathogenic *Pseudomonas* strains in the bodies of patients with AD. Future research should focus on further understanding the role of specific bacterial clusters in the gut microbiome in the pathogenesis of AD [58]. A study where the young WT mice received a gut microbiota transplant from old AD mice has shown that this intervention significantly impaired the recovery from a traumatic brain injury. The study has also shown increased activation of microglia and macrophages and reduced motor recovery. In addition, there was a higher relative count of *Muribaculum* bacteria and a decrease in *Lactobacillus johnsonii* in WT mice transplanted with a microbiome derived from old AD mice. Another study confirms that the microflora derived from AD mice has a significant effect on the deterioration of the neurological response [59]. 

### 2.2. Microbiome in Huntington’s Disease

#### 2.2.1. Trinucleotide Repeat Expansion Disorders

The expansion of microsatellite repeats is the cause of several neurodegenerative diseases. They are usually caused by replication errors such as polymerase dissociation or arrest, or sliding of the 5′ and or 3′ ends of the Okazaki fragment, which results in the formation of a hairpin structure [60,61]. Neurodegenerative diseases that are classified as trinucleotide repeat expansion disorders (TREDs) are caused by the repetition of the CNG sequence (where N is one of the 4 nucleotides) in certain genes. These disorders can further be subclassified as PolyQ (where the repeated sequence CAG encodes glutamine), like Huntington’s disease (HD), and Spinocerebellar Ataxia types 1, 2, 3, 6, 7, 12, 17, and non-PolyQ (where other triplets are repeated), like myotonic dystrophy (DM) or Friedreich’s ataxia (FRDA) [62,63].

#### 2.2.2. Huntington’s Disease Etiology

Huntington’s disease is a rare disorder of the CNS. It affects 5–10 in 100,000 people [64]. It is the most common disorder in Europe and USA, and the least in Asia [65,66,67]. 

HD symptoms include uncontrolled body movements, weight loss, facial grimaces, psychological disorders, personality changes, and apathy. First non-specific symptoms can start 10 years before full manifestation of HD, which usually occurs between 35 and 40 years of age. The disease can also affect juveniles, but it is extremely rare in patients under the age of 10 and over the age of 70. The life expectancy after first symptoms is 15–20 years, with the most common causes of death being aspiration pneumonia, heart disease, and suicide [68,69,70]. The mutation that causes HD is located in the first exon of the *HTT* gene and is inherited in an autosomal dominant manner. In healthy individuals, the first exon contains between 10 and 35 CAG repeats, and the disease severity varies depending on the number of repeats: 27–35 repeats do not cause the disease but increase the probability of HD manifestation in progeny; 36–38 repeats cause the disease with incomplete penetrance; and more than 39 repeats cause the disease with complete penetrance, where the first symptoms occur in patients at the age of 40–55. More than 60 repeats cause the juvenile form, where the first symptoms occur before the age of 21 [71]. This specific mutation in *HTT* leads to the expression of mutant HTT (mHTT) protein, which tends to form intracellular insoluble aggregates that are the pathologic hallmark of HD [72]. The longer the polyQ repeats, the more aggregates it forms. In the brain, the disease pathology is linked to neuronal loss in the striatum, which is responsible for control of motor functions and the reward center. Medium spiny neurons make up the structure of the striatum, and these cells are mainly affected by pathogenic mHTT aggregates, which lead to neuronal loss and secondary gliosis. The other hallmarks of HD pathology are weight loss, gastritis, esophagitis, and nutritional deficiencies, all of which point to a strong link with dysfunction of the digestive tract. mHTT has been found to be expressed in the majority of tissues, including the gastrointestinal tract. Interestingly, studies performed on mouse models have shown that mHTT forms aggregates in the enteric nervous system even before neurological and motoric symptoms appear. It has also been reported that HD affects the functions of the gastrointestinal (GI) system through impaired gut motility, diarrhea, and malabsorption of food, and even influences the gut anatomy by reducing mucosal thickness and villus length, as well as the loss of various neuropeptides that stimulate or inhibit gut motility [73]. There are also pathological changes in gene transcription—mHTT aggregates have been found to interact with several proteins involved in various transcriptional pathways. They have been found to interact with specificity protein 1 (SP1), CREB-binding protein (CBP), peroxisome proliferator-activated receptor-γ coactivator 1α (PGC1α), Nuclear factor κ light-chain-enhancer of activated B cells (NF-κB), and Repressor element 1 (RE1)-silencing transcription factor (REST) [74]. Altered transcription in HD is also linked to mitochondrial dysfunction. Diminished transcription of PGC1α negatively impacts energy metabolism and mitochondrial biogenesis. The mHTT has also been found to have a strong association with the translocase of mitochondrial inner membrane 23 (TIM23) complex, which impairs protein import and disrupts mitochondrial function [74,75,76]. 

#### 2.2.3. Immunoprofiling of Huntington’s Disease

Chronic inflammation is a hallmark of HD. Inflammatory responses predate motor and psychiatric symptoms, suggesting that chronic inflammation contributes to disease progression. mHTT is highly expressed in immune cells, and its aggregates have been found to have a proinflammatory effect [77]. Even in premanifest patients, peripheral inflammation is characterized by elevated plasma levels of IL-6, and IL-8, IL-4, IL-10, and TNF-α levels rise as the disease progresses. The increase of both IL-6 and IL-8 in the early stages suggests, that it is the innate immunity that drives the initial immunopathology in HD. Indeed, monocytes, macrophages, and microglia isolated from HD patients were found to be hyperreactive to stimulation [78]. The mHTT has been found to drive up the release of IL-6 by upregulating the NF-κB pathway in mice [79]. Interestingly, a study has shown that the presence of mHTT does not directly impact the function of T cells, as their frequencies and functions did not differ from healthy controls [80]. 

Central inflammation in HD is characterized by chronic activation of microglia and astrocytes. Microglia are the primary mediators of neuroinflammation and in their activated state they release proinflammatory cytokines, such as IL-6, IL-1β, and TNF-α, as well as cytotoxic factors, such as reactive oxygen species (ROS), nitric oxide (NO) and QUIN. Prolonged microglial activation can lead to chronic neuroinflammation and tissue damage [81]. The number of activated microglial cells has been shown to positively correlate with the degree of neuronal loss in the striatum and cortex [82]. It has also been found that activation of microglia is present in very early stages of disease prior to the onset of symptoms [83]. Unlike microglia, the activation of astrocytes occurs in later stages of disease, when neurodegeneration is already present [81]. Reactive astrocytes can contribute to the proinflammatory environment through the production of pro-inflammatory cytokines, such as IL-12 and TNF-α; however, they can also contribute to neuroprotection by expressing anti-inflammatory cytokines, such as IL-10 and TGF-β [84]. 

Several studies have found a link between T helper 17 (Th17) cells and immunopathology in HD. In premanifest gene expansion carriers, it has been found that Th17.1 cells are activated while the number of Tregs is diminished. IL-17 is a proinflammatory cytokine that plays a role in communication between immune cells and tissue. In animal models, it has been shown to interact with endothelial cells, which induces the breakdown of the blood-brain barrier. The presence of IL-17 in cerebrospinal fluid (CSF) activates microglia, astrocytes, and oligodendrocytes, causing neuroinflammation. Early therapeutic intervention targeting Th17 cells might be beneficial and delay the onset of symptoms [85]. 

#### 2.2.4. Microbiome in Huntington’s Disease


**HD—mouse model studies**


There are several commercially available mouse models of HD. They differ in the genetic background, the structure of the transgene, and the disease phenotype. The most commonly used lines are R6/1 and R6/2, which are characterized by early symptoms and rapid progression of the disease, compared to the BACHD line. The BACHD mouse model shows the first symptoms of the disease between 2 and 6 months of age, but their severity appears after about a year. The BACHD line shows somatic stability in embryos [86]. 

Studying the microbiome is an increasingly emerging trend in HD research. One study showed an impact of the transplantation of a microbiome derived from WT mice into a mouse model of HD on its phenotype. The results show that especially the females responded positively to this procedure, as improvements in cognitive function have been observed in animals suffering from HD. The same study proved the ineffectiveness of this approach in males. Researchers speculated that the possible reasons for that phenomenon might be more extensive changes in structure, instability in the gut microbiome and the imbalance in acetate immune profiles [87]. In order to characterize the gut microbiome in a mouse model of HD, 16S RNA sequencing was performed. The research was carried out on R6/1 mice. Sequencing results revealed significant differences in the composition of the microbiome. Furthermore, the amount of water in the feces of HD mice at 12 weeks of age was significantly changed. Most notably, there was an increase in *Bacteroidetes* and a proportional decrease in *Firmicutes*. Interestingly, an increase in microbiome diversity was also observed in HD males compared to WT control mice, but these differences were not observed in females. The changes coincided with an increased food intake and a simultaneous decrease in body weight [88]. It has been proven that PD is characterized by a decrease in the expression of TJ proteins, which under physiological conditions maintain the integrity of the intestinal barrier [89]. Björkqvist and coauthors evaluated whether the same mechanism is responsible for the pathologies occurring in another mouse model of HD (R6/2). The results showed a significant decrease in body weight and body length in these mice. They were also accompanied by a decrease in colon length compared to WT mice, but TJ protein levels showed no statistically significant changes between groups. Moreover, along with the observed changes, differences in the composition of the gut microbiota were also found in the R6/2 mice. Increased amounts of *Bacteroidetes* and *Proteobacteria* and decreased amounts of *Firmicutes,* relative to levels maintained in the control group were demonstrated [90]. A very interesting and detailed study was performed by Gubert et al. They focused on comparing the study group (R6/1 mouse line), which consisted of 3 subgroups: animals with standard living conditions, mice with additional environmental enrichment, and groups of animals with increased physical activity, with WT mice as controls. The results indicated a possible modulation of the gut microbiome by the environment. Therapeutic effects on psychomotor symptoms and the brain have been reported in groups of animals with an enriched environment and greater activity compared to the control group. Changes in the composition of the microbiome at the level of orders such as *Bacteroidales*, *Lachnospirales,* and *Oscillospirales* have also been demonstrated. The results obtained in this experiment show higher alpha diversity for all HD mice compared to WT mice. There was no difference in food intake, but there was a previously expected decrease in body weight in the HD mice compared to the control group. Increased water intake by animals from the test groups was shown, which was associated with the increase in alpha diversity. With the aging of the HD animals, increased fecal excretion was noted. Post-mortem analysis showed a statistically significant decrease in the brain weight of HD mice. There were also significant differences between males and females. The brain weight of females was lower in the group of mice with standard living conditions. Based on the study of the concentration of SCFAs and branched chain fatty acids (BCFAs) in the feces, an attempt was made to check what role these metabolites may play in living condition changes. Male mice from the group with increased physical activity were characterized by a decrease in the concentration of butyrate and valerate. There was no correlation between the concentration of substances, such as acetate and propionate, and the living conditions, genotype, or gender. Statistically significant differences were found between HD and WT mice in the alpha diversity index. The test groups showed increased alpha diversity indices in contrast to the control group. The results of the beta diversity analysis showed differences between the sexes of the animals. Certain orders of microbial bacteria have been identified as those that play the greatest role in microbiome changes under different animal housing conditions. These include the orders *Bacteroidales*, *Lachnospirales,* and *Oscillospirales* [91]. 

Early pathological features associated with HD are molecular deficits in myelination and progressive neurodegeneration. Experiments conducted on germ-free (GF) animals suggested that there is a two-way communication between the microbiome, gut, and brain [11,92]. Research conducted on the BACHD mouse model was intended to answer the question of what impact the microbiome has on myelin plasticity and oligodendrocyte dynamics. The experiment compared GF, specific pathogen-free (SPF), and WT mice. Animals of both sexes were used in the experiment. Analysis of myelin in the corpus callosum revealed changes in myelin thickness in BACHD GF mice compared to SPF mice, while no intergroup changes were observed in WT mice. However, significant differences in myelin density were noted in all groups compared to WT SPF mice. In the GF conditions, a reduced level of myelin-associated proteins, such as myelin basic protein (MBP), proteolipid protein (PLP), and Ermin (Ermn), and a lower number of mature oligodendrocytes in the prefrontal cortex were observed compared to the SPF conditions, regardless of the mouse genotype. Slight differences in family and genus were also observed in the commensal bacteria of the gut microbiome in the BACHD and WT groups maintained under SPF conditions. However, the differences were not statistically significant. Researchers concluded that the *HTT* mutation in BACHD mice does not cause profound disturbances in the intestinal microflora, and thus plasticity defects are not associated with disturbances in the structure of the microbiome. Analysis of the brain structures of GF animals showed that then environment had a greater effect on the myelination caliber of callosum axons in BACHD animals compared to WT controls, while a possible distribution of myelin plaques was observed in both genotypes. The axons of mice maintained under GF conditions were characterized by a reduced diameter and a lower g-ratios, which could suggest thicker myelin. Examination of the myelin membranes, however, showed that the observed features may have been due to the decompaction of the laminae and not an increase in their number. A similar trend of increased periodicity, suggesting decompaction, was also observed in BACHD mice under SPF conditions compared to WT, prompting the conclusion that the *HTT* mutation in BACHD animals causes this pathology. Supportive is the observation of a trend towards lower levels of the cortical myelin-associated proteins MBP and PLP, which play a key role in myelin compaction. The researchers did not observe significant changes in the gut bacterial community. Slight disparities were observed in BACHD mice at 3 and 6 months of age compared to WT mice, with reduced numbers of *Prevotella* and *Bacteroides* at the genus level and part of the *Bacteroidetes* type [93]. More reports indicate the importance of the intestinal microbiome in the communication between the digestive system and the brain and its impact on the pathologies of neurodegenerative diseases. Subsequent studies involved shotgun sequencing of the gut microbiome from R6/1 mice, aged 4–12 weeks (from early adolescent to adult stages). Metabolomic analyses, in addition to those performed on fecal samples, were also performed on blood plasma collected from 12-week-old animals. The results showed an upregulation of bacterial gene expression, which may indicate potential early effects of the HTT protein mutation in the gut. In addition, mice at 12 weeks of age were found to have disturbed gut microbiome function. In particular, the researchers’ attention was drawn to the increase in the butanoate metabolic pathway, which leads to increased production of SCFA playing a protective role. This increase was not observed when analyzing plasma from 12-week-old mice. Statistical analysis of the results obtained in metagenomic and metabolomic studies allowed for the observation of a negative correlation of several species of *Bacteroides* with ATP and pipecolic acid in plasma. During the experiment, feces were collected at five different time points. No statistically significant differences in the composition of the microbiome were observed when comparing the mice from the study group and the control WT group. The dominance of two phyla, *Bacteroidetes* and *Firmicutes,* was observed, followed by the *Proteobacteria*, *Actinobacteria,* and *Verrucomicrobia* phyla. It was determined that at the family level, the most numerous group was *Lachnospiraceae*, followed by similar numbers in the groups of *Bacteroidaceae*, *Porphyromonadaceae*, *Prevotellaceae,* and *Clostridiaceae*. No statistically significant differences were found between bacterial families at any timepoint when comparing WT mice. At 12 weeks of age, which corresponds to the timepoint before the onset of overt motor symptoms in HD mice, differences in 30 bacterial species were observed between HD and WT mice. These included *Clostridium mt 5*, *Treponema phagedenis*, *Clostridium leptum CAG: 27*, *Desulfatirhabdium butyrativorans*, *Plasmodium chabaudi*, *Defulfuribacillus alkaliarsenatis*, *Plasmodium yoelii,* and *Chlamydia abortus*. No differences in the abundance of butyrate producers such as *Roseburia intestinalis*, *Clostridium symbiosum,* and *Eubacterium rectale* were found when comparing samples from HD and WT mice [94].**HD—human studies**

Studies were also performed on a diverse group of people suffering from HD. Participants were clinically characterized using a battery of cognitive tests, and 16S RNA sequencing was performed on stool samples. The study involved healthy individuals (control group; n = 36) and carriers of the expanded mutated gene (n = 42). Nineteen of them were previously diagnosed with HD, and the rest were pre-symptomatic. The groups were matched by gender and age. Microbiome evaluation showed differences between the control group and the study group in the composition of the microbial community (beta diversity) as well as significantly lower species richness (alpha diversity). The results of the sequencing analysis show statistically significant differences at the phylum level (differences apply only to the group of men) in *Euryarchaeota*, *Firmicutes,* and *Verrucomicrobia*. Further changes were also observed at the family level, including: *Acidaminococcaceae*, *Akkermansiaceae*, *Bacteroidaceae*, *Bifidobacteriaceae*, *Christensenellaceae*, *Clostridiaceae*, *Coriobacteriaceae*, *Eggerthellaceae*, *Enterobacteriaceae*, *Erysipelotrichaceae*, *Flavobacteriaceae*, *Lachnospiraceae*, *Methanobacteriaceae*, *Peptococcaceae*, *Peptostreptococcaceae*, and *Rikenellaceae*, concerned only men. No significant changes at the phylum and family levels were observed in women. The obtained results confirmed the researchers’ assumptions and showed changes in the composition of the microbiome between the test and control groups. In addition, the observations made provide evidence that the composition of the intestinal microbiome affects the cognitive abilities of patients. However, the results obtained in this study should be interpreted with caution. According to the authors, the study and control groups were too small to make adequate statistical analyses. Nevertheless, the information provided is essential for further research [95]. Another study conducted on patients suffering from HD indicates a correlation between changes in the composition of the gut microbiome and the immune response. The study included 33 HD patients and 33 healthy individuals; the groups were matched in terms of sex and age. In addition to assessing the fecal microflora in terms of microbial richness, structure, and diversity of abundance of individual taxa, IFN-γ, IL-1β, IL-2, IL-4, IL-6, IL-8, IL-10, IL-12p70, IL-13, and TNF-α concentrations in patients’ plasma were measured. The results obtained in both experiments were correlated with each other to find connotations between them. It was shown that HD patients were distinguished by increased richness and altered microbiome structure. The analysis showed that the higher number of *Intestinimonas* bacteria is positively correlated with the Total Functional Capacity score (measured in HD patients to evaluate disease progression). It is also positively correlated with the level of the anti-inflammatory cytokine IL-4. The study also showed that the genus *Bilophila* is negatively correlated with pro-inflammatory IL-6 levels. In addition, negative correlations between *Oscillibacter*, *Gemmier,* and IL-6; *Clostridium XVIII*, TNF-α and IL-8; and positive correlations between *Porphyromonas* and IL-4, IL-10, and IL-13 were also noted. The results obtained in these experiments clearly indicate the relationship between the composition of the intestinal microbiome and the immune response in HD patients [96]. 

All results described in the paragraph are summarized in Table 1. 

## 3. Discussion and Future Prospects

Increasing advancement in research on neurodegenerative diseases indicates that these pathologies are very complex processes with often forgotten microbiome- and immune-related components. The publications and studies mentioned in this review present evidence for the relationship between neurodegenerative diseases, mainly HD, and the intestinal microbiome. So far, the focus has been on understanding the pathology of the disease based on molecular biomarkers, which hopefully could effectively contribute to the development of future therapies [97]. Recent studies on the effect of the intestinal microbiome and its metabolites also pave the way for new branches in the field of HD. Microbial metabolites have the potential to modulate the pathogenesis of HD. SCFAs can influence the immune system and ameliorate inflammation, both in the CNS and the peripheral nerves. Studies on mouse models that were supplemented with sodium butyrate showed a beneficial effect on their motor skills, mitochondrial and transcriptional dysfunction [28,29]. This suggests that therapeutic interventions promoting butyrate production by patients’ microbiota have the potential to ameliorate disease symptoms. However, there are still many open questions regarding the bacteria inhabiting healthy and diseased digestive systems. The results of research involving microbiota carried out so far are still not entirely conclusive due to microbiome complexity and numerous contributing factors. Therefore, there is still a long way to go to fully understand the communication in the gut—brain axis, including in pathological conditions like HD.

Moreover, the microbiome results are not always consistent. The large amount of data generated in experiments is hard to compile, and one needs to be attentive when analyzing and drawing conclusions based on it. Insufficiently known taxonomies of species inhabiting the intestines and inaccurate and non-standardized terminology related to the subject of the microbiome are often misleading and generate mistakes when classifying individual bacteria into appropriate classes, groups, or families. Furthermore, the choice of mouse model, its strain, sex, or age is essential in the studies concerning the microbiome. For example, two studies in a mouse model of HD confirm an increase in *Bacteriodetes* and a decrease in *Firmicutes* [88,90]. The first one was carried out on the R6/1 line, and the second on the R6/2 mouse model. Additionally, the study conducted on another HD model contradicts these results. At 3 months of age, BACHD mice exhibit the opposite trend of increased *Firmicutes* and decreased *Bacteriodetes*. Interestingly, re-analysis on 6-month-old mice showed the opposite, which rather confirms the results of the previous two studies [93]. The presented results display certain consistency, despite the use of different models, but only when using older mice from the BACHD line. It can be assumed that the microbiome diversity changes in the same fashion as organisms mature. It is also worth noting that some of the results show statistical significance only in the group of males, both in animal and human studies [88,95]. On the other hand, only female mice showed a positive reaction to the transplant of a healthy microbiome [87]. These findings also indicate the effect of female hormones on microbiome composition. In the study conducted by the Hannan group, the body weight of WT and HD mice differed significantly, as HD mice lost weight with age. This could be due to differences in the composition of the microbiome and the level of food absorption, which is inversely proportional to body weight [98]. Increased thirst was also noted, possibly due to xerostomia, which both patients and HD mice suffer from, or hypothalamic degeneration, which is associated with increased thirst [99]. Interestingly, increased water intake by the animals did not change the water content of the feces. The reason could be the microbiological environment in the intestines. This result may suggest a very precise regulation of water absorption [100]. Some of the cited studies indicate an increased level of alpha diversity compared to other groups [88,91,96]. A higher level of this index is believed to indicate a healthier and more resilient microbial environment [101]. Studies in other models of neurodegenerative diseases, such as AD and PD, have also linked movement deficits with lower levels of alpha diversity in patients compared to controls [102,103,104]. Human HD studies have shown lower [95] and higher [96] values of alpha diversity in CAG repeat overexpressors compared to healthy controls. Recent extensive meta-analyses have found no associations between alpha diversity and neurological disorders, particularly in PD and MS [105]. Interestingly, there are also studies that prove that increased diversity does not always correlate with better patient conditions [106,107]. According to Coyte et al., a decrease in the stability of the microbiome environment may also result in higher alpha diversity [108]. Research also shows that the alpha level of diversity may also be related to diet, body weight, and gastrointestinal physiology [109]. 

Another essential factor that should be considered when conducting experiments related to the microbiome and neurodegeneration in humans is the environment. Each of the mentioned experiments was performed under slightly different conditions, especially in humans. Environmental changes are noticeable among the participants of a project, despite the fact that the control group was chosen from the close family members of the patients [96]. The composition of the gut microbiome is also influenced by various factors, such as physical activity [110,111]. The difference in this respect between healthy and disease-affected individuals certainly existed during the project. This proves how difficult it is to compose appropriate groups in experiments assuming the study of the microbiome. In addition to differences in physical activity, each person has different nutritional preferences, which certainly influence the composition of the microbiome and are a burden for bioinformaticians to be leveled in statistical analyses [112,113]. In addition, the quoted research was performed on distinct continents, which results in diametrically different environmental conditions such as climate or local food accessibility that affect diet [114]. Sampling for testing is an extremely important point in the whole experiment. Typically, the collected samples are snap frozen to eliminate the adverse effect of air on aerophobic bacteria in the samples. In both of these experiments [95,96], the samples were obtained in a different way, and the patients were responsible for collecting and delivering the samples to the laboratory, which might have affected the composition of the microbiome in the samples. Conducting research on mouse models can be better standardized and reproducible by applying a specific sampling and storage protocol. Collection should be as quick as possible, with a caution not to contaminate the sample with other DNA or with bacteria residing on fur.

Animal experiments also have the advantage of breeding in more standardized conditions, typically SPF, though the microbiome may vary slightly. On the other hand, the place of origin of the animal, lineage, strain, age, disease model, maintenance method, or even environmental enrichment in the cages are all aspects that should be considered when studying the microbiome. Mice are also known to be coprophages to reabsorb essential nutrients such as vitamins; thus, when housing a few mice in the same cage, one should consider the natural microbiome transfer between them and dodge the “cage effect” [91]. Additionally, all existing mouse models of HD differ from each other by the dynamics of disease progression or the degree of interference in the animal’s genome [115,116]. At this point, it is worth considering at what age and on what model such tests should be carried out. The studies we quoted were based on various models and were carried out on animals of different ages. As with human studies, comparing results obtained in mouse experiments is equally problematic, although the experiments were more standardized.

Animal models of HD provide us with tools to study the mammalian microbiome and its possible implications for disease progression in a highly controlled environment. Most studies presented in this review used R6/1 or R6/2 models, which are well established for HD; however, they are characterized by early onset, rapid disease progression, and premature death. As previously mentioned, in humans, the symptoms of HD occur well into adulthood, at 35–40 years of age, with continuous progression for the next 10–15 years, which points to a need for other models with slower disease progression, such as YAC128, Hu128/21, or BACHD. Aging is also closely linked to changes in microbiome composition, so these models might be more applicable for long term studies of changes in microbiome composition and possible dietary or therapeutic interventions that might better translate to humans. There was only one study utilizing the BACHD model that showed pronounced differences in microbiota composition at different ages [93]. Long-term studies on both pre- and post-symptomatic animals are important for a better understanding of the microbiome and HD pathology, but they also have the unique ability to find the most suitable timepoints for therapeutic interventions. Using these models might also be relevant in fecal microbiota transfer studies, as the R6/2 model used by Gubert and colleagues has shown that the engraftment was unsuccessful in male mice [87]. Using models with slower disease progression might provide the researchers with a variety of timepoints and disease phenotypes to choose from, which might impact the success of the microbiota transplant.

There is also a fruit fly model of HD (FL-HD) that exhibits similar symptoms such as motor deficits, mHTT aggregates, disrupted gene expression, and dysbiosis in the gut. The *Drosophila* microbiome is, however, much less complex than the mammalian microbiome, which can help in analyzing single species and their impact on dysbiosis. A study conducted on female fruit flies has found that gut colonization by *E. coli* worsened the HD symptoms, as there was an increase in aggregate buildup and earlier death. A therapy using crocin was used in *Drosophila* with beneficial effects. This therapy ameliorated motor deficits and extended the lifespan, but what is more interesting is that it provided resistance to *E. coli* colonization and had positive effects on the microbiome [117]. Crocin is a carotenoid exhibiting anti-inflammatory, antioxidant, and neuroprotective properties. Crocin, or its major byproduct, crocetin, has been suggested to act in the gut and modulate the gut microbiome. Another study has shown that oral administration of crocin was beneficial for cerebral ischemic/reperfusion (I/R) injuries in rats, while the intravenous route of administration was not. It suggests that the therapeutic effects are mediated through the gut microbiota [118]. As such, crocin might provide beneficial effects in HD, ameliorating inflammation, oxidative stress, and gut dysbiosis, which makes it a promising target for further studies. 

Interestingly, a few studies have found that prion infection can also lead to dysbiosis and significant changes in microbial metabolites. The microbial richness (alpha diversity) was higher in healthy controls, and the microbiome structure was significantly different between healthy and infected groups. Prion diseases are linked to neuroinflammation, and while the mechanism underlying the gut dysbiosis in this type of disease is not well understood, it is nonetheless an interesting topic to further examine the relationship between the gut and the brain [119,120].

According to the latest research, taking pro- and pre-biotics can help with nervous system diseases. So far, the effect of taking these substances on the progression of HD has not been proven, but it has been studied in other neurodegenerative diseases. There are several studies confirming the psychophysiological effect of prebiotics on the body. Chitosan oligosaccharide (COS) has been shown to have a positive effect on cognitive deficits in a rat model of AD by reducing oxidative stress and neuroinflammatory responses [121]. In studies on amyotrophic lateral sclerosis, it was proven that the use of galactooligosaccharides (GOS) reduced the activation of microglia and astrocytes and caused less death of motor neurons [122]. Other studies conducted in a mouse model of PD showed that long-term intake of probiotics resulted in a neuroprotective effect on dopaminergic neurons, effectively counteracting motor disorders in animals [123]. Unfortunately, few similar studies have been conducted in humans so far. The examples of research cited above prove that the use of products containing both pro- and prebiotic bacterial strains could act as an effective supporting therapy in the treatment of neurodegenerative diseases. Perhaps in the future, effective and personalized drugs based solely on these compounds will be developed.

## Figures and Tables

**Table 1 ijms-24-04477-t001:** Summarized results of microbiome studies performed in HD mouse models and HD patients. The table shows bacterial species changed in HD, ↑ signifies increase, ↓ signifies decrease. C – Class, O – Order, F – Family, G – Genus.

Host	Phylum	Class/Order	Family/Genus	Alpha Diversity	Beta Diversity	Source
Mouse R6/1	*Bacteroidetes* ↑ *Firmicutes* ↓			males↑	ns	[88]
Mouse R6/2	*Bacteroidetes* ↑ *Proteobacteria* ↑ *Firmicutes* ↓		**G***Bacteroides* ↑ **G** *Parabacteroides* ↑ **G** *Lactobacillus* ↑ **G** *Coprobacillus* ↑ **G** *Enterobacteriaceae* ↑	ns		[90]
Mouse R6/1		**O** *Bacteroidales* **O** *Lachnospirales* **O** *Oscillospirales*		↑	differed	[91]
Mouse BACHD 3 months old	*Bacteroidetes* ↓ *Firmicutes* ↑		**F***Bacteroidaceae* ↓ **F** *Anaeroplasmataceae* ↓ **G** *Prevotella* ↓ **G** *Bacteroides* ↓ **G** *Oscillospira* ↑ **G** *Adlercreutzia* ↑	ns		[93]
Mouse BACHD6 months old	*Bacteroidetes* ↑ *Firmicutes* ↓		**F***Mogibacteriaceae* ↓	ns		[93]
Human Males	*Firmicutes* ↓ *Euryarchaeota Verrucomicrobia*		**F***Lachnospiraceae* ↓ **F** *Akkermansiaceae* ↓ **F** *Acidaminococcaceae* **F** *Akkermansiaceae* **F** *Bacteroidaceae* **F** *Bifidobacteriaceae* **F** *Christensenellaceae* **F** *Clostridiaceae* **F** *Coriobacteriaceae* **F** *Eggerthellaceae* **F** *Enterobacteriaceae* **F** *Erysipelotrichaceae* **F** *Flavobacteriaceae* **F** *Lachnospiraceae* **F** *Methanobacteriaceae* **F** *Peptococcaceae* **F** *Peptostreptococcaceae* **F** *Rikenellaceae*	↓	differed	[95]
Human	*Actinobacteria* ↑	**C***Deltaproteobacteria* ↑ **C** *Actinobacteria* ↑ **O** *Desulfovibrionales* ↑	**F***Oxalobacteraceae* ↑ **F** *Lactobacillaceae* ↑ **F** *Desulfovibrionaceae* ↑ **G** *Clostridium XVIII* ↓ **G** *Intestinimonas* ↑ **G** *Bilophila* ↑ **G** *Lactobacillus* ↑ **G** *Oscillibacter* ↑ **G** *Gemmiger* ↑ **G** *Dialister* ↑	↑	differed	[96]

## Data Availability

Not applicable.

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
