# Peer review of "What the Gut Tells the Brain—Is There a Link between Microbiota and Huntington’s Disease?"

_ijms, 2023, doi:10.3390/ijms24054477_

Round 1
Reviewer 1 Report
The manuscript has an interesting and new topic that worths being investigated.
The following comments could be mentioned:
1. It is not clear what are the keywords of the article, some are separated by "," or ";". The keyword MeSh on Demand could be used for better indexing the article
2. What does the drawing on the first page represent? There is no legend.
3. The manuscript contains many informations, some of them outside the scope of the review. For this reason, the Introduction chapter could be shortened; the subchapter "2.1. Parkinson’s Disease and Alzheimer’s Disease" could be removed; the Discussion chapter could be shortened also.
4. The font in Table 1 could be increased for enhanced clarity, especially since it contains the main data form the literature relating microbiota to Huntington disease
5. The Discussion chapter is interesting but it contains manly limitations of all these studies. It should manly resume to the direct evidences found in relationship microbiota and Huntington disease.
6. There are many references cited. A few references should be revised, and could be placed in the manuscript according to the template.
Author Response
- It is not clear what are the keywords of the article, some are separated by "," or ";". The keyword MeSh on Demand could be used for better indexing the article
Thank you for pointing out the issues regarding the keywords. We have computed the abstract by MeSh on Demand, as recommended and decided to follow some of the program suggestions. We replaced some of the keywords as well as included the new ones. We added the following keywords: “Neurodegeneration”; “Dysbiosis“ and instead of “Microbiota” we added “Gastrointestinal Microbiome”. We think modified keywords better reflect the article’s scope.
- What does the drawing on the first page represent? There is no legend.
Thank you for this comment. Indeed the drawing misses a legend. The first drawing represents a graphical abstract, which according to MDPI’s instructions does not require a legend. However, we agree it would be beneficial to the readers to include the label that it is the graphical abstract and we added it below the graphical abstract. We leave it for the editor’s consideration whether it could be included or not in a final version of the manuscript.
- The manuscript contains many informations, some of them outside the scope of the review. For this reason, the Introduction chapter could be shortened; the subchapter "2.1. Parkinson’s Disease and Alzheimer’s Disease" could be removed; the Discussion chapter could be shortened also.
Thank you for these suggestions. We removed lines 63-66 from the introduction. We think that indeed this historical outline is irrelevant for the context.
We have partly shortened the paragraphs concerning Alzheimer’s and Parkinson’s diseases, however to a limited extend. The idea behind including AD and PD was to relate the changes found in microbiome of HD to these well-studied and most-prevalent neurodegenerative diseases. Furthermore, it gives a broadened overview on the spectra of neurological disorders and microbiome.
We have removed sentences:
“Considering recent research on the gut microbiota in PD, no specific bacteria has been found directly related to this disease.”
“In a healthy body, they are responsible for maintaining the integrity of the intestinal barrier and shaping the innate immunity of the intestinal mucosa.”
We also would like to acknowledge the reviewer that we include a short paragraph in discussion on prion disease and microbiota, along with the other reviewer’s comment.
“Interestingly, a few studies have found that prion infection can also lead to dysbiosis and significant changes in microbial metabolites. The microbial richness (alpha diversity) was higher in healthy controls and the microbiome structure was significantly different between healthy and infected groups. Prion diseases are linked to neuroinflammation and while the mechanism underlying the gut dysbiosis in this type of diseases is not well understood, it is nonetheless an interesting topic to further examine the relationship between the gut and the brain. “
https://doi.org/10.1016/j.nbd.2019.104704 , 10.3390/pathogens10070887
- The font in Table 1 could be increased for enhanced clarity, especially since it contains the main data form the literature relating microbiota to Huntington disease
Thank you for this comment. We adjusted the summary table to be more readable. We widened it and increased the font. However, we would like the editorial office to advise on this issue as we have used the template for that.
- The Discussion chapter is interesting but it contains manly limitations of all these studies. It should manly resume to the direct evidences found in relationship microbiota and Huntington disease.
Limitations of the studies were meant to show how challenging it is to interpret and draw conclusions based on the experimental data due to their high variability. Although, we made an attempt to propose a guideline for those willing to study microbiome in animal models. Direct evidences are summarized in the table 1 for clarity, and we felt there was no need to specifically enlist those direct evidences again in discussion.
- There are many references cited. A few references should be revised, and could be placed in the manuscript according to the template.
Thank you for this suggestion. We did a thorough check on the references and removed references 4-5, namely:
[4] Van Leeuwenhoek, A. “An Abstract of a Letter from Antonie van Leeuwenhoek, Sep. 12, 1683. about Animals in the Scrurf of the Teeth.” Philos. Trans. R. Soc. Lond 1684.14: 568-574.
[5] Ursell, L.K.; Metcalf, J.L.; Parfrey, L.W.; Knight, R. Defining the Human Microbiome. Nutrition Reviews 2012, 70, S38–S44, doi:10.1111/j.1753-4887.2012.00493.x.
Reviewer 2 Report
Authors in such review described the alteration of microbioma in NDs, in particular by considering the correlation of such alteration with HD. The review is well written and also figures are clear.
I have only a comment.
It should be nice whether authors add some comments regarding microbiome&prion diseases (neuroinflammation, transmissibility)
Author Response
Authors in such review described the alteration of microbioma in NDs, in particular by considering the correlation of such alteration with HD. The review is well written and also figures are clear.
I have only a comment.
- It should be nice whether authors add some comments regarding microbiome&prion diseases (neuroinflammation, transmissibility)
We thank reviewer for his valuable suggestion. In response to the comment we added a short paragraph concerning the gut microbiome and prion diseases in discussion part. It reads:
“Interestingly, a few studies have found that prion infection can also lead to dysbiosis and significant changes in microbial metabolites. The microbial richness (alpha diversity) was higher in healthy controls and the microbiome structure was significantly different between healthy and infected groups. Prion diseases are linked to neuroinflammation and while the mechanism underlying the gut dysbiosis in this type of diseases is not well understood, it is nonetheless an interesting topic to further examine the relationship between the gut and the brain.“
https://doi.org/10.1016/j.nbd.2019.104704 , 10.3390/pathogens10070887
Round 2
Reviewer 1 Report
The authors responded to all issues previously addressed. All changes made improved the quality of the manuscript. I have no further comments.